# Nanobodies combined with DNA-PAINT super-resolution reveal a staggered titin nanoarchitecture in flight muscles

**Florian Schueder[1,2†], Pierre Mangeol[3†], Eunice HoYee Chan[3], Renate Rees[4], Jürgen Schünemann[4], Ralf Jungmann[1,2]\*, Dirk Görlich[4]\*, Frank Schnorrer[3]\***

[1]Faculty of Physics and Center for Nanoscience, Ludwig Maximilian University, Munich, Germany; [2]Max Planck Institute of Biochemistry, Martinsried, Germany; [3]Aix Marseille University, CNRS, IBDM, Turing Centre for Living Systems, Marseille, France; [4]Max Planck Institute for Multidisciplinary Sciences, Göttingen, Germany

**\*For correspondence:**
jungmann@biochem.mpg.de (RJ);
goerlich@mpinat.mpg.de (DG);
frank.schnorrer@univ-amu.fr (FS)

[†]These authors contributed equally to this work

**Competing interest:** The authors declare that no competing interests exist.

**Abstract** Sarcomeres are the force-producing units of all striated muscles. Their nanoarchitecture critically depends on the large titin protein, which in vertebrates spans from the sarcomeric Z-disc to the M-band and hence links actin and myosin filaments stably together. This ensures sarcomeric integrity and determines the length of vertebrate sarcomeres. However, the instructive role of titins for sarcomeric architecture outside of vertebrates is not as well understood. Here, we used a series of nanobodies, the *Drosophila* titin nanobody toolbox, recognising specific domains of the two *Drosophila* titin homologs Sallimus and Projectin to determine their precise location in intact flight muscles. By combining nanobodies with DNA-PAINT super-resolution microscopy, we found that, similar to vertebrate titin, Sallimus bridges across the flight muscle I-band, whereas Projectin is located at the beginning of the A-band. Interestingly, the ends of both proteins overlap at the I-band/A-band border, revealing a staggered organisation of the two *Drosophila* titin homologs. This architecture may help to stably anchor Sallimus at the myosin filament and hence ensure efficient force transduction during flight.

## Editor's evaluation

This landmark study combines two advanced technologies, namely, nanobodies and DNA-PAINT, to define the position of several subdomains of the two fly titin homologs in adult flight muscles. Their results provide compelling evidence that Sallimus can bridge the Z-disk with the beginning of the myosin A-band. Furthermore, their results convincingly establish that Sallimus and Projectin partially overlap at the beginning of the A-band. The work should appeal to people interested in muscle biology and more generally to people interested in providing high-resolution images of long proteins.

## Introduction

Skeletal and heart muscles produce forces that power body movements and fluid flow in animals. These forces are produced by conserved macromolecular machines called sarcomeres. Sarcomeres are organised into long periodic chains called myofibrils that mechanically span the entire muscle fibre length and thus sarcomere contraction results in muscle contraction (*Gautel, 2011*; *Huxley, 1969*; *Lemke and Schnorrer, 2017*).

The sarcomeric architecture is conserved in striated muscles across animals. Sarcomeres are bordered by two Z-discs, which anchor the plus ends of parallel actin filaments. These extend towards

**eLife digest** From ants to humans, the muscles that set an organism in motion are formed of bundles of fiber-like cells which can shorten and lengthen at will. At the microscopic level, changes in muscle cell lengths are underpinned by contractile filaments formed of multiple repeats of a basic unit, known as the sarcomere. Each unit is bookended by intricate 'Z-discs' and features an 'M-band' in its center.

Three protein types give a sarcomere its ability to shorten and expand at will: two types of filaments (myosin and actin), which can slide on one another; and a spring-like molecule known as titin, which ensures that the unit does not fall apart by mechanically connecting myosin and actin. More specifically, actin filaments are anchored to the Z-discs and extend towards the M-band, while myosin filaments are centered around the M-band and extend towards the Z-discs. As myosin and actin slide alongside each other, the overlap between the two types of filaments increases or decreases and the whole unit changes its length.

In vertebrates, one gigantic molecule of titin spans from the Z-disc to the M-band, linking together actin and myosin filaments and determining the length of the sarcomere. In insects and other invertebrates, however, this single molecule is replaced by two titin proteins known as Projectin and Sallimus. Understanding how these titins work together remains unclear and difficult to study. Traditional approaches are unable to precisely label titin in an environment teaming with other molecules, and they cannot offer the nanometer resolution required to dissect sarcomere organization.

As a response, Schueder, Mangeol et al. combined super-resolution microscopy and a new toolbox of labelling molecules known as nanobodies to track the position of Sallimus and Projectin in the flight muscles of fruit flies. These experiments revealed that the two proteins are arranged in tandem along the length of the sarcomere, forming a structure that measures about 350 nm. Sallimus is anchored in the Z-disc and it runs alongside actin until it reaches the end of a myosin filament; there, it overlaps with Projectin for about 10 nm. Projectin then stretches for 250 nm along the length of the beginning myosin filament.

These findings confirm the importance of titin in dictating the length of a sarcomere; they suggest that, in invertebrates, this role is split between two proteins, each possibly ruling over a section of the sarcomere. In addition, the work by Schueder, Mangeol et al. demonstrate the value of combining nanobodies and super-resolution microscopy to study complex structures in tissues.

the centrally located bipolar myosin filaments that are cross-linked at the M-band of the sarcomere. In vertebrate sarcomeres, actin and myosin filaments are mechanically linked by the connecting filament built by the gigantic titin protein, whose N-terminus is anchored to α-actinin at the Z-disc while its C-terminus is embedded within the sarcomeric M-band. Thus, titin spans as a linear protein across half a sarcomere in vertebrate muscle (*Gautel and Djinović-Carugo, 2016*; *Lange et al., 2006*; *Linke, 2018*; *Squire et al., 2005*). This stereotypic sarcomere architecture results in a defined sarcomere length (distance between two neighbouring Z-discs), which is about 3 μm in relaxed human skeletal muscle (*Ehler and Gautel, 2008*; *Llewellyn et al., 2008*; *Regev et al., 2011*), and is responsible for the typical striated appearance of skeletal muscles.

The defined sarcomeric architecture sparked the 'titin ruler hypothesis', proposing that the long titin protein rules sarcomere length in vertebrate muscles (*Tskhovrebova and Trinick, 2012*; *Tskhovrebova and Trinick, 2017*). Recently, this hypothesis has been strongly supported by in vivo genetic evidence. Deletion of parts of titin's flexible I-band or its stiff A-band regions in mouse skeletal muscle resulted in a shortening of the sarcomeric I-band or A-band, respectively (*Brynnel et al., 2018*; *Tonino et al., 2017*). Furthermore, recent evidence substantiated that titin is the main sarcomeric component responsible for the passive tension of the muscle, suggesting that mechanical tension present in relaxed muscle is stretching titin into its extended conformation (*Li et al., 2020*; *Linke, 2018*; *Rivas-Pardo et al., 2020*; *Swist et al., 2020*). Thus, titin mechanically links actin and myosin filaments together and is responsible for establishing and maintaining sarcomeric architecture in vertebrate striated muscle.

Striated muscle architecture is not restricted to vertebrates but is conserved in insects and nematodes. However, in contrast to vertebrates, titin's role in *Drosophila* and *Caenorhabditis elegans*

appears to be split in two proteins, one containing the flexible I-band features and the other the stiff A-band features of titin (*Flaherty et al., 2002*; *Loreau et al., 2023*; *Porto et al., 2021*; *Tskhovrebova and Trinick, 2003*). Surprisingly, the sarcomere length in flies and worms is still stereotypic for the respective muscle fibre type. In *Drosophila,* the sarcomere length is about 3.5 µm for indirect flight muscles and about 8 µm for larval body wall muscles. To date, it is unclear how sarcomere length in these muscles is determined. Furthermore, it is unknown how the titin homologs are precisely organised within the sarcomere and whether they contribute to sarcomere length regulation in insect muscle.

We aimed to address the questions of how invertebrate titin homologs instruct sarcomere architecture and whether the titin nanoarchitecture would be consistent with a ruler function mechanically linking actin to myosin at a defined distance, as proposed for vertebrates. A first step to answer these important questions is to determine the exact positions of the titin homologs within the sarcomere.

Here, we chose the *Drosophila* indirect flight muscles to determine the precise topology of the two *Drosophila* homologs Sallimus (Sls) and Projectin (Proj). We selected key domains at different locations within Sls and Projectin, against which we recently raised specific nanobodies (*Loreau et al., 2023*). We applied single and dual-colour DNA-PAINT super-resolution microscopy to intact flight muscle specimens, which determined the precise architecture of Sls and Projectin in the flight muscle sarcomere. Interestingly, we found that Sls but not Projectin extends from the Z-disc to the myosin filament. The end of Sls overlaps with the beginning of Projectin, which projects further along the myosin filament. This staggered organisation of the two *Drosophila* titin homologs may explain how high mechanical tension can be stably transmitted across the sarcomere and how sarcomere length can be ruled without the presence of a single protein linking the Z-disc to the M-band as observed in vertebrates.

## Results
### *Drosophila* titin domain organisation and flight muscle isoforms

*Drosophila* indirect flight muscles (called flight muscles in the remainder of the article) are stiff muscles that oscillate at high frequency to power flight (*Dickinson, 2006*; *Pringle, 1981*; *Schönbauer et al., 2011*). The majority of this stiffness is due to Sls in flight muscles (*Kulke et al., 2001*). To achieve this high stiffness, a large part of the flexible spring domains encoded in both titin gene homologs *sls* and *bent* (*bt*; protein name: Projectin) are skipped by alternative splicing (*Ayme-Southgate et al., 2005*; *Bullard et al., 2005*; *Burkart et al., 2007*; *Spletter et al., 2015*). Older work had suggested that the most prominent Sls flight muscle isoform (also called Kettin) uses an alternative poly-A site terminating the protein after Sls-immunoglobulin (Ig) domain 35 (*Bullard et al., 2005*; *Burkart et al., 2007*). However, more recent systematic transcriptomics and splice-site annotation data from dissected flight muscles, as well as expression of large genomic Sls-GFP tagged transgenes, showed that the usage of this early poly-A site is largely restricted to leg muscles and hardly present in flight muscles (*Spletter et al., 2015*; *Spletter et al., 2018*). To identify the most prominent Sls and Projectin protein isoforms in mature flight muscles, we carefully reanalysed the published transcriptomics and splice data (*Spletter et al., 2015*; *Spletter et al., 2018*). We verified that in both genes the flexible PEVK spring domains are largely spliced out in adult flight muscles; however, their more 3'-located exons are present at least in some longer isoforms (*Figure 1—figure supplement 1A and B*). This predicts an Sls isoform containing the C-terminal five fibronectin (Fn) domains and a Projectin isoform containing a long stretch of Ig-Fn super-repeats and a kinase domain close to its C-terminus being present in flight muscles (*Figure 1—figure supplement 1A and B*).

### Sallimus and Projectin nanobodies in flight muscles

In order to verify the expression and to determine the precise location of the different Sls domains in adult flight muscle sarcomeres, we selected three different regions in Sls, against which we recently generated nanobodies: Sls-Ig13/14, Sls-Ig49/50, and Sls-Ig51/Fn2, the first being relatively close to the N-terminus, the other two being close to the C-terminus of the Sls flight muscle isoform (*Loreau et al., 2023*; *Figure 1A*). Similarly, we selected two regions in Projectin close to its N-terminus (Proj-Ig5-8 and Proj-Fn1/2) and two regions close to its C-terminus (Proj-Ig27-Fn35 and Proj-kinase domain)

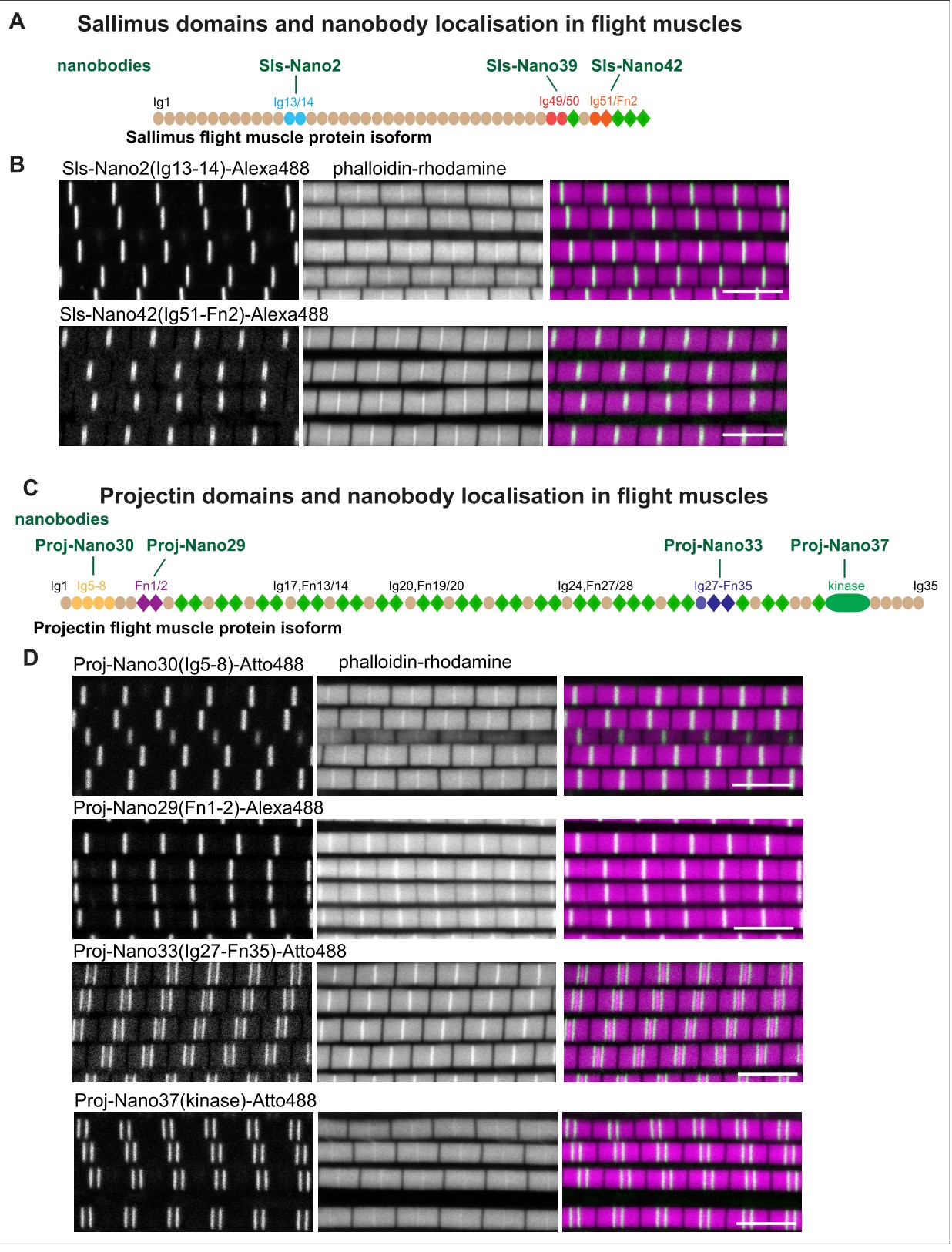

**Figure 1.** *Drosophila* titin domain organisation and nanobodies. (**A, C**) Sallimus (**A**) and Projectin (**C**) flight muscle protein isoforms with the domains recognised by the used nanobodies highlighted in different colours. (**B, D**) Single confocal sections of flight muscle sarcomeres from adult hemi-thoraces stained for actin with phalloidin (magenta) and the indicated anti-Sls or anti-Projectin nanobodies directly coupled to Alexa488 or Atto488 (green). The Z-disc is revealed by the prominent actin signal. Scale bars 5 μm.

*Figure 1 continued on next page*

*Figure 1 continued*

The online version of this article includes the following figure supplement(s) for figure 1:

**Figure supplement 1.** Flight muscle titin isoforms.

---

(*Figure 1C*). The generation of these nanobodies as well as their specificity was documented in an accompanying manuscript (*Loreau et al., 2023*).

In flight muscles, both N-terminal anti-Sls nanobodies, namely, Sls-Nano2 (binding Sls-Ig13/14) as well as C-terminal anti-Sls nanobodies, namely, Sls-Nano42 (binding to SlsIg51/52), result in single bands at the Z-discs when coupled to fluorescent dyes and observed with confocal microscopy (*Figure 1B*). Similarly, N-terminal anti-Projectin nanobodies, here Proj-Nano30 (recognising Proj-Ig5-8) and Proj-Nano29 (binding Proj-Fn1-2), each result in one band at the Z-disc (*Figure 1D*). In contrast, nanobodies against C-terminally located Projectin domains, here Proj-Nano33 (binding Proj-Ig27-Fn35) and Proj-Nano37 (binding the Projectin kinase domain), result in two bands at large distances from the Z-disc (*Figure 1D*). These data demonstrate that Projectin is present in an extended conformation, and since the flight muscle I-band extends less than 100 nm from the Z-disc (*Burkart et al., 2007*; *Kronert et al., 2018*; *Loison et al., 2018*; *Reedy and Beall, 1993*; *Szikora et al., 2020*), a large part of Projectin is present along the myosin filament. However, the diffraction-limited spatial resolution of a confocal microscope (about 250 nm) is not sufficient to precisely localise Sls and Projectin domains close to the Z-disc. Hence, higher spatial resolution is necessary to determine the precise architecture of Sls and Projectin within the flight muscle sarcomere.

## DNA-PAINT super-resolution imaging of entire flight muscles

To resolve the relative localisation of Sls and Projectin, we turned our attention to super-resolution imaging with DNA-PAINT as it enables imaging at particularly high spatial resolution (*Jungmann et al., 2014*; *Lelek et al., 2021*; *Schnitzbauer et al., 2017*). For DNA-PAINT, nanobodies binding the protein epitope of interest need to be site-specifically conjugated to either one- or two single-stranded DNA molecules. Previously, DNA-oligos for PAINT were either coupled to antibodies via biotin-streptavidin (*Jungmann et al., 2014*), which is a 66 kDa tetramer and thus relatively large, or more frequently by click chemistry (*Fabricius et al., 2018*; *Schnitzbauer et al., 2017*), which comes with a number of potential disadvantages, such as a bulky hydrophobic coupling group and an initial lysine modification that might destroy the paratope. Instead, we used maleimide-coupling through ectopic cysteines at the N- and C-terminus of the nanobody (*Pleiner et al., 2018*; *Pleiner et al., 2015*), which allows a simpler workflow, analogous to direct fluorophore coupling, and protects the antigen-binding site from undesired modifications.

In contrast to fluorophore-maleimides, maleimide-activated oligonucleotides are not commercially available. However, as described in the 'Methods' in detail, they are straightforward to synthesise from a 5′-amino-modified oligo and a bifunctional maleimide-NHS (N-hydroxysuccinimide) cross-linker (*Figure 2*). The NHS group forms an amide bond with the 5′-amino group of the oligo under reaction conditions that leave the amino groups of the DNA bases non-reactive. The maleimide-activated oligo is then reacted with the nanobody that still contains its His14-SUMO or His14-NEDD8 tag (*Frey and Görlich, 2014*). The resulting conjugate is purified by binding to an Ni(II) chelate matrix (whereby non-conjugated oligo remains in the non-bound fraction) and followed by elution of nanobody-oligo conjugate with a tag-cleaving protease (*Figure 2*). Hence, these oligo-coupled nanobodies remain similarly small as the fluorescently coupled nanobodies and thus are ideal for effective super-resolution imaging using DNA-PAINT.

In DNA-PAINT, the necessary target blinking for localisation-based super-resolution reconstruction is achieved by the transient binding of a dye-labelled single-stranded DNA 'imager' strand to their target-bound complement ('docking' strands, *Figure 3A*). As imager strands are continuously replenished from solution and binding times are controllable over a wide range, a large number of photons can be detected from a single binding event, thus enabling unprecedented sub-5 nm spatial resolutions (*Dai et al., 2016*; *Schnitzbauer et al., 2017*).

We aimed to apply DNA-PAINT to flight muscle tissue, using hemi-thoraces of adult flies, to minimise artefacts that might be introduced by cutting out individual myofibrils. To prepare hemi-thoraces, we fixed thoraces in paraformaldehyde and then bisected them with a sharp microtome knife (*Figure 3—figure supplement 1A*, see 'Methods' for details). Then, we incubated the hemi-thoraces

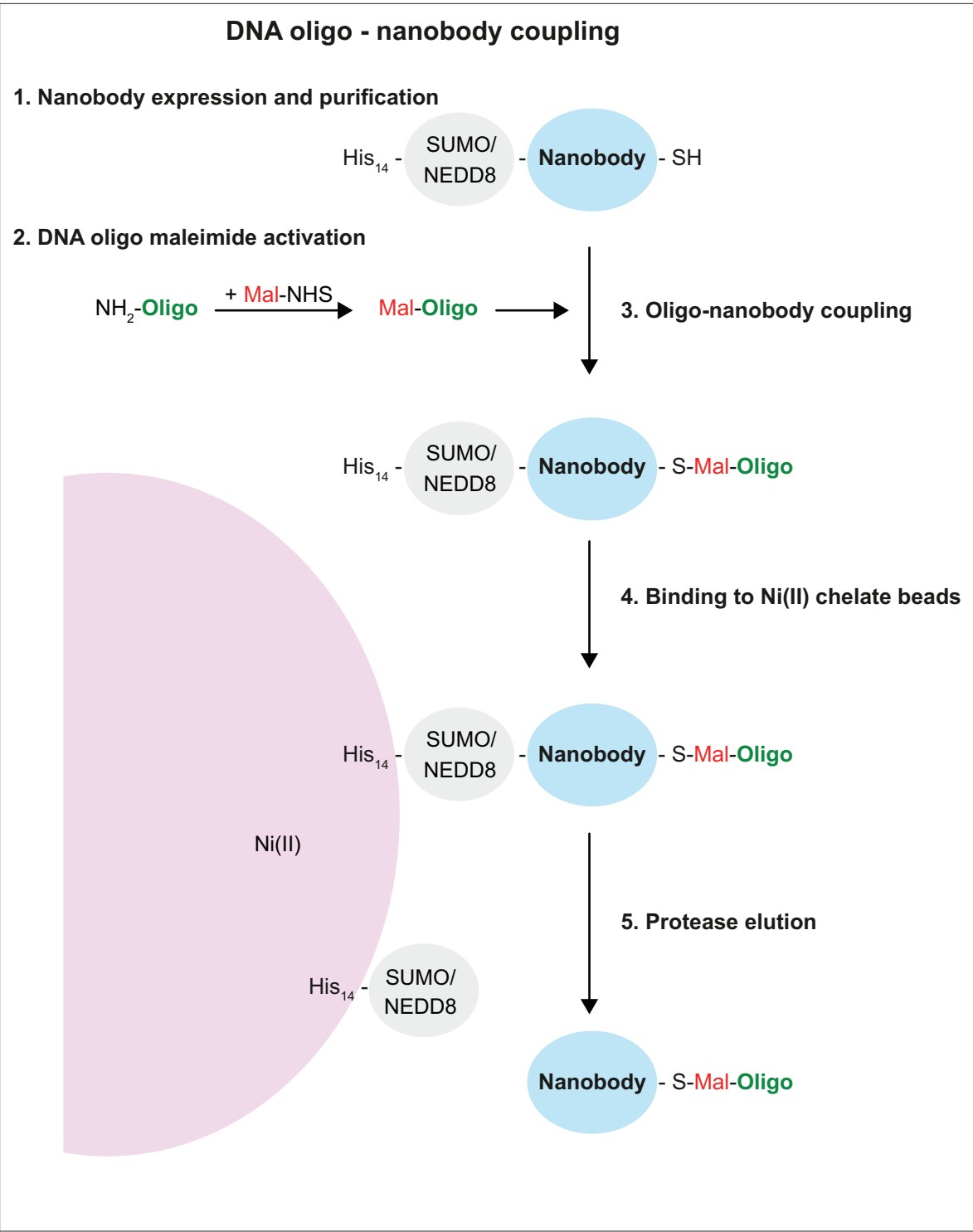

**Figure 2.** Single-strand DNA oligonucleotide-nanobody coupling. Schematic representation of the five steps from nanobody purification, oligo activation and coupling, purification, and elution of the oligo-coupled nanobody. See 'Results' and 'Methods' sections for details.

with oligo-coupled nanobodies and mounted them for imaging. Hemi-thoraces are very large, with a length of about 1 mm and a thickness of about 300 μm. To mount them as close as possible to the coverslip, we developed an imaging chamber that contains the imaging buffer surrounded by spacers thick enough to slightly press the flight muscles against the coverslip (*Figure 3—figure supplement 1A*, see 'Methods' for details). This enabled DNA-PAINT imaging with total internal reflection (TIRF).

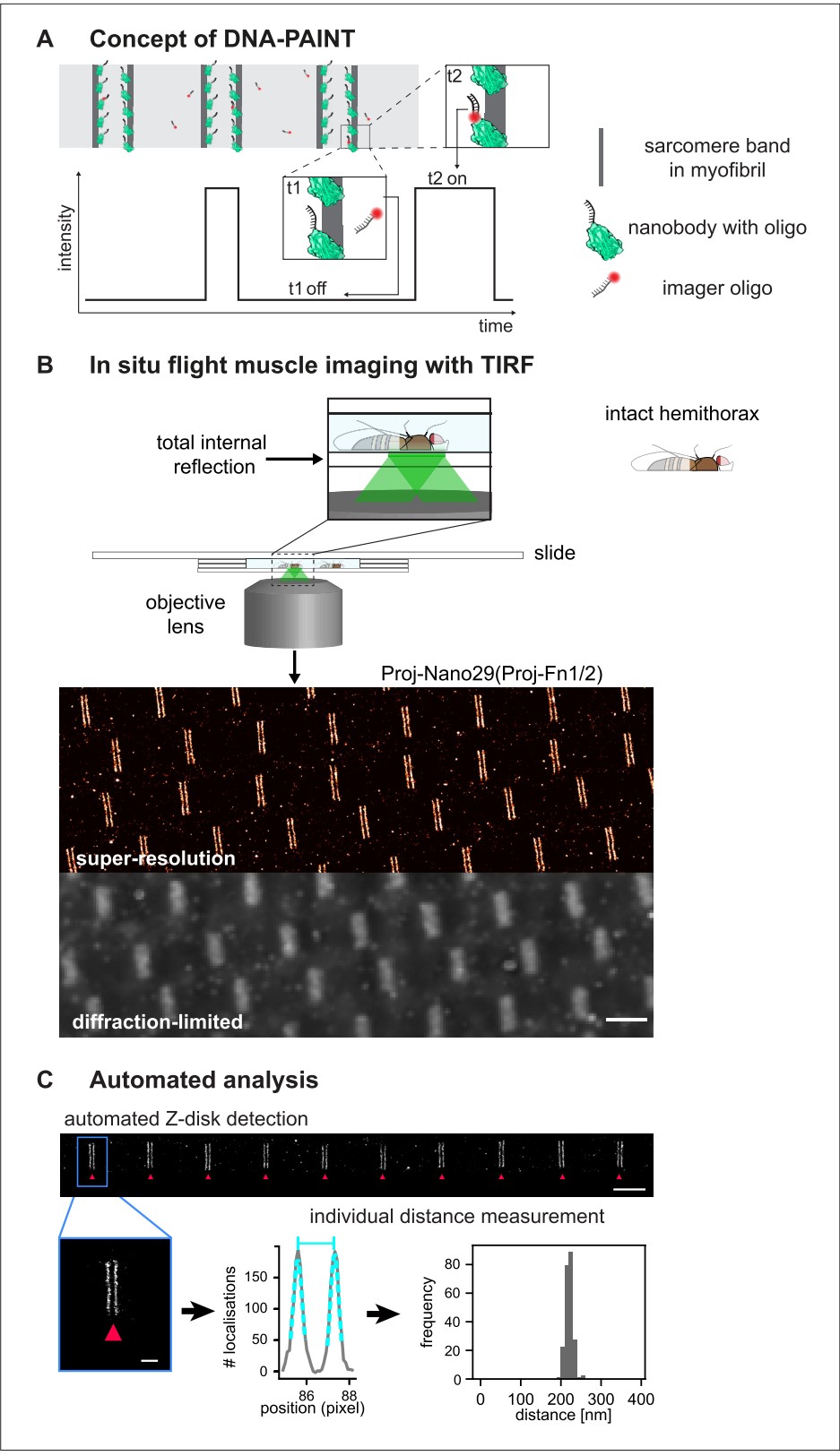

**Figure 3.** *Drosophila* flight muscle DNA-PAINT imaging and automated extraction of sarcomeric protein domain positions. (**A**) Concept of DNA-PAINT imaging of sarcomeres labelled with an oligo-conjugated nanobody. Binding of the imager oligo to one nanobody results in a strong, detectable intensity burst (t2, blink). (**B**) Schematic of a mounted intact *Drosophila* hemi-thorax in a DNA-PAINT imaging chamber enabling total internal reflection

*Figure 3 continued on next page*

*Figure 3 continued*

(TIRF) illumination. Comparison of the diffraction-limited and the super-resolved result illustrated in one hemi-thorax labelled with Proj-Nano29. Note that the super-resolved image can readily resolve the two bands flanking each Z-disc. Scale bar 2 μm. (**C**) Automated image analysis for individual Z-discs detection (see *Figure 3—figure supplement 2* and 'Methods' for details). Individual bands are detected automatically and their centre position is obtained using a Gaussian fit (bottom centre). The distance between the centre of bands for tens of sarcomeres from a single hemi-thorax is then reported in a histogram (bottom right). Scale bar 2 μm (top) and 0.5 μm (bottom).

The online version of this article includes the following source data and figure supplement(s) for figure 3:

**Source data 1.** Distances between Proj-Nano29 bands in shown example.

**Figure supplement 1.** Sample preparation and data processing.

**Figure supplement 2.** Data analysis workflow.

We imaged for 30 min per sample and obtained about 15,000 frames at an imaging rate of 10 Hz. For image reconstruction and post-processing, we used the established Picasso software (*Schnitzbauer et al., 2017*; *Figure 3B*, *Figure 3—figure supplement 1B*; see 'Methods' for details). This enabled us to resolve the two bands flanking a Z-disc with ease, which cannot be resolved in the diffraction-limited image (*Figure 3B*).

To further refine the precision of determining the epitope positions, we have developed an image-processing pipeline that relies on an interactive selection of well-stained myofibrils in the volume of TIRF excitation (*Figure 3—figure supplement 2*). Next, we removed localisations arising from multiple binding events by filtering based on specific localisation parameters (see 'Methods' for details). Furthermore, we automatically detected the individual sarcomeric Z-discs and the respective flanking bands of the stained Sls or Projectin epitopes for all selected myofibrils. We applied a Gaussian fit to each band and determined their centre positions within the sarcomere with nanometric accuracy (*Figure 3C*, *Figure 3—figure supplement 2*). This results in an accurate location of the measured bands for each of the epitopes in every analysed sarcomere. Hence, we do not need to average across many sarcomeres to precisely localise the Sls or Projectin epitopes (*Figure 3C*). In conclusion, our method allows detecting individual differences in sarcomeric band positions in each sarcomere investigated down to the nanometre-scale.

## Positions of Sallimus and Projectin domains within intact flight muscle at the nanometric scale

To precisely determine the location of Sallimus and Projectin, we applied our DNA-PAINT imaging pipeline of flight muscles to the entire Sls and Projectin nanobody toolbox (*Loreau et al., 2023*). In most cases, we co-stained with two nanobodies that are spaced sufficiently apart to detect the expected four bands centred around the Z-disc, even when using only a single imaging colour (*Figure 4A*). This allowed us to resolve the positions of Sls-Nano2 (Sls-Ig13/14) located close to the N-terminus of Sallimus and also Sls-Nano39 (Sls-Ig49/50) close to its C-terminus, which we could combine with distantly located anti-Projectin nanobodies (*Figure 4A*). Similarly, we imaged the N-terminally located Proj-Nano29 (Proj-Fn1/2) and Proj-Nano30 (Proj-Ig5-8), which we combined with one of the C-terminally located Proj-Nano33 (Ig27-Fn35), Proj-Nano35, or Proj-Nano37 (both Projectin kinase domain). This enabled us to locate the exact position of the different Projectin domains in sarcomeres (*Figure 4A*). Interestingly, all the analysed epitopes localise in similarly sharp bands in each of the sarcomeres, suggesting a very precisely aligned architecture of Sls and Projectin. The result that nanobodies recognising Sls-Ig13/14 localise 50 nm away from the centre of the Z-disc is compatible with the very N-terminus of Sls being located at the centre of the Z-disc because the additional twelve N-terminal 12 Ig domains are likely to span 48 nm (longest dimension of an Ig domain = 4 nm), whereas the N-terminus of Projectin is located around 100 nm away from the Z-disc (Proj-Ig5-8 and Proj-Fn1/2) (*Figure 4A*) and hence cannot be anchored directly at the Z-disc. Our distance measurements are very reproducible between samples, as shown for the different samples stained with Sls-Nano2 in *Figure 4A* as well as between different nanobodies that recognise neighbouring domains as shown for Proj-Nano29 and Proj-Nano30. Hence, the combination of DNA-PAINT with the oligo-labelled nanobodies works very reliably to image sarcomeres in intact flight muscle tissue.

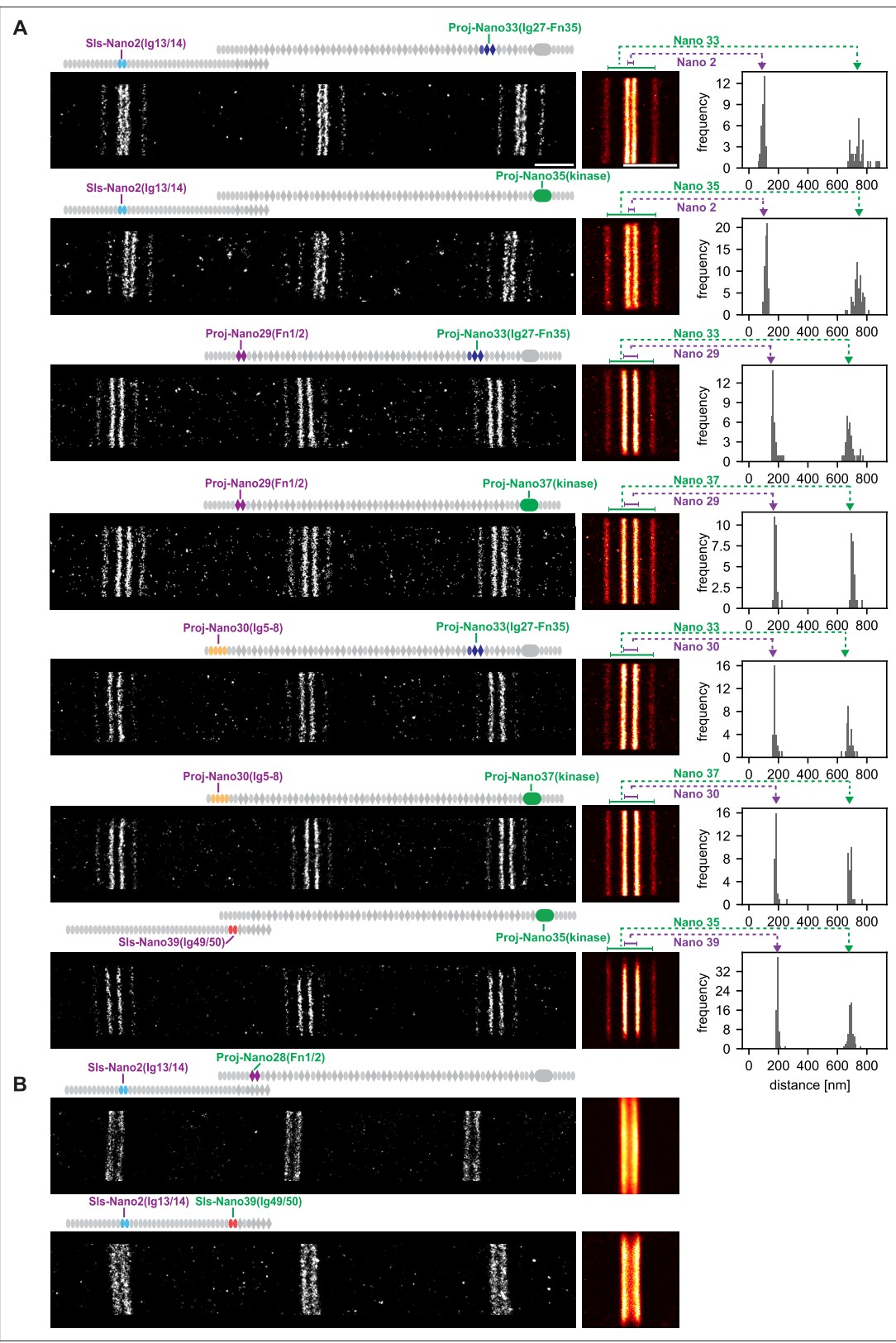

**Figure 4.** Single-colour DNA-PAINT imaging of Sallimus (Sls) and Projectin domains. (**A**) Left: representative DNA-PAINT images of myofibrils stained with two different anti-Sls or anti-Projectin nanobodies labelling two epitopes and imaged with the same fluorescent imager oligo. The different Sls or Projectin nanobody combinations are indicated above each image. Middle: pseudo-coloured sum image centred around Z-discs resulting from one hemi-thorax. Right: histogram of distances between bands centred around Z-discs with the respective nanobody combinations indicated in green or

*Figure 4 continued on next page*

*Figure 4 continued*

magenta. The frequency (y-axis) refers to the number of Z-discs present in the respective length bins. Note that four bands can be readily distinguished for all shown nanobody combinations. The number of Z-discs scored in the shown samples are from top to bottom: 34, 59, 35, 25, 28, 28, and 63 (see *Figure 4—source data 1* for the individual measurements). (**B**) Similar representations as in (**A**). However, the positions of neighbouring Sls or Projectin epitopes cannot be resolved in a single colour. Scale bar 1 µm.

The online version of this article includes the following source data for figure 4:

**Source data 1.** Band distances and imaging conditions os samples shown in *Figure 4*.

However, this powerful single-colour imaging method has limitations: it fails to resolve two different epitopes into distinct bands if these epitopes are located too close together to unambiguously assign each blinking event to one particular nanobody. Thus, Sls-Nano2 (Sls-Ig13/14) and Sls-Nano39 (Sls-Ig49/50) or Sls-Nano2 (Sls-Ig13/14) and Proj-Nano28 (Proj-Fn1/2) cannot be imaged together in the same sarcomere with a single colour (*Figure 4B*). However, quantifying the exact positions of two closely located titin domains in the same sarcomere is critical as the relative length of the flexible titin molecules may vary in individual sarcomeres. Hence, it would be important to determine the positions of two different Sls domains in the same sarcomere to unambiguously conclude about Sls length or the relative arrangement of Sls and Projectin protein domains.

## Two-colour DNA-PAINT reveals a staggered organisation of Sls and Proj

To simultaneously determine the exact positions of two epitopes, we have labelled two nanobodies each with two different oligonucleotides and imaged them with two differently labelled imager oligos in parallel to perform two-colour DNA-PAINT (see 'Methods'). Multiplexed imaging enabled us to determine the positions of Sls-Ig13/14 (using Sls-Nano2) and Sls-Ig51-Fn2 (using Sls-Nano42) in the same sarcomere (*Figure 5A*). Our results verified that Sls-Ig13/14 is localised about 50 nm away from the centre of the Z-disc and that Sls-Ig51/Fn2 is about 50 nm farther towards the middle of the sarcomere (*Figure 5A*, *Figure 5—figure supplement 1*). Since the I-band of flight muscles is less than 100 nm from the Z-disc (*Burkart et al., 2007*; *Kronert et al., 2018*; *Loison et al., 2018*; *Reedy and Beall, 1993*; *Szikora et al., 2020*), this strongly suggests that Sls is bridging across the entire sarcomeric I-band with its N-terminus anchored within the Z-disc and its C-terminal end reaching the myosin filament. Thus, Sls could mechanically link the Z-disc to the myosin filament in the flight muscles, similar to the long vertebrate titin.

Since we found that the N-terminus of Projectin is also about 100 nm away from the Z-disc and thus located at the beginning of the thick filament (*Figure 4A*), we wanted to further investigate the precise orientation of the Projectin N-terminal domains. To do so, we performed two-colour DNA-PAINT to localise Proj-Ig5/8 (with Proj-Nano30) and Pro-Fn1/2 (with Proj-Nano29) in the same sarcomere and found an average distance between the two epitopes of about 25 nm, with Proj-Ig5-8 being always closer to the Z-disc relative to Proj-Fn1/2 (*Figure 5B*, *Figure 5—figure supplement 1*). Consistently, the more C-terminally located Proj-Ig27-Fn35 epitope is located far into the myosin filament that begins at 100 nm (*Szikora et al., 2020*), being 350 nm away from the Z-disc (*Figure 5B*, *Figure 5—figure supplement 1*). This strongly suggests that the N-terminal part of Projectin is arranged in an extended, likely linear conformation reaching from the myosin filament into the I-band and thus running in parallel to the C-terminal domains of Sls.

These findings raised an enticing hypothesis: do the extended Sls and Projectin proteins overlap at the I-band/A-band interface? To investigate this hypothesis, we performed two-colour DNA-PAINT using two pairs of nanobodies: Sls-Nano39, recognising Sls-49/50, combined with Proj-Nano29, binding Proj-Fn1/2 and Sls-Nano42, recognising Sls-Ig51-Fn2, combined with Proj-Nano30, binding Proj-Ig5/8. Interestingly, we found that in all sarcomeres measured, the Proj-Nano29 is about 15 nm farther from the Z-disc than Sls-Nano39, whereas, in 42 out of 45 sarcomeres investigated, Proj-Nano30 is on average 7–8 nm closer to the Z-disc than Sls-Nano42 (*Figure 5C*, *Figure 5—figure supplement 1*). Hence, these data revealed an interesting staggered organisation of the two overlapping ends of the linearly extended Sallimus and Projectin proteins in flight muscles.

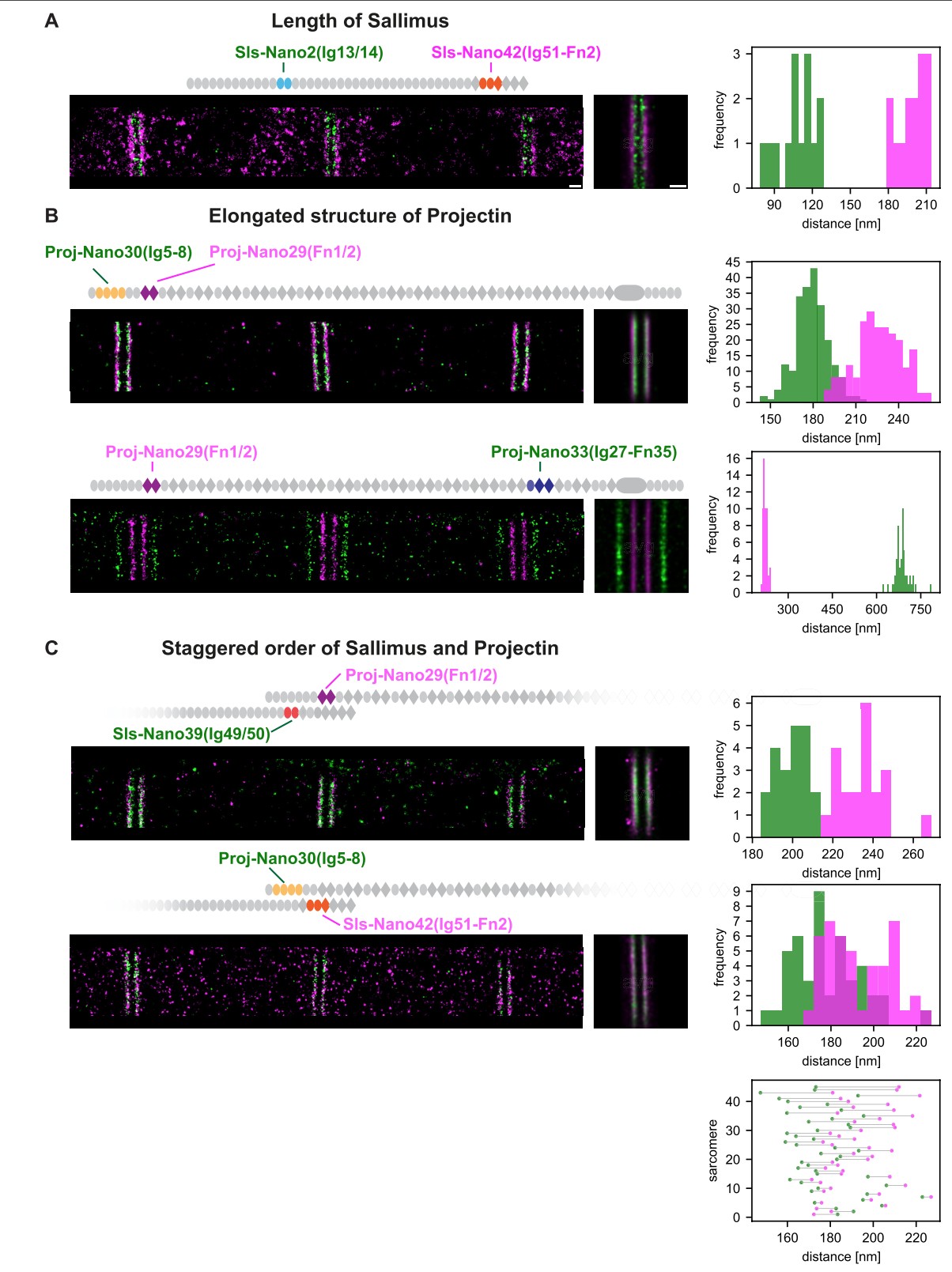

**Figure 5.** Dual-colour DNA-PAINT imaging reveals staggered order of Sallimus (Sls) and Projectin. (**A**) Left: representative DNA-PAINT image of a myofibril stained with two nanobodies labelling Sls-Ig13/14 (Sls-Nano2) and Sls-Ig51/Fn2 epitopes (Sls-Nano42). Middle: sum image centred around Z-discs resulting from one hemi-thorax. Right: histogram of distances between bands centred around Z-discs (Sls-Ig13/14 in green, Sls-Ig51/Fn2 in magenta). The frequency (y-axis) refers to the number of Z-discs present in the respective length bins (number of Z-discs scored: 14, see **Figure 5—**

*Figure 5 continued on next page*

*Figure 5 continued*

source data 1 for the respective measurements). (**B**). Top: representative DNA-PAINT image of a myofibril stained with two nanobodies labelling Proj-Ig5-8 (Proj-Nano30) and Proj-Fn1/2 (Proj-Nano29) epitopes, sum image, and histograms of distances between bands (Proj-Ig5-8 in green, Proj-Fn1-2 in magenta, number of Z-discs scored: 219). Bottom: representative myofibril stained for Proj-Fn1/2 (Proj-Nano29) and Proj-Ig27-Fn35 (Proj-Nano33) epitopes, sum image, and histogram of distances between bands centred around Z-discs (Proj-Fn1/2 in magenta, Proj-Ig27-Fn35 in green, number of Z-discs scored: 52) (**C**). Top: representative DNA-PAINT image of a myofibril stained with two nanobodies labelling SlsIg49/50 (Sls-Nano39) and Proj-Fn1/2 (Proj-Nano29) epitopes, sum image, and histogram of distances between bands centred around Z-discs (Sls-Ig49/50 in green, Proj-Fn1/2 in magenta, number of Z-discs scored: 21). Bottom: same as top for Sls-Ig51/Fn2 (Sls-Nano42) and Proj-Ig5-8 (Proj-Nano30) epitopes, sum image, histogram of distances, and plot showing the epitope positions from the Z-discs in the individual sarcomeres analysed (bottom right, Sls-Ig51/Fn2 in magenta, Proj-Ig5-8 in green, number of Z-discs scored: 45). Note that in 42 of 45 cases the Proj-Ig5-8 (green) is closer to the Z-disc than Sls-Ig51/Fn2 (magenta). Scale bar 250 nm.

The online version of this article includes the following source data and figure supplement(s) for figure 5:

**Source data 1.** Band distances and imaging conditions for all samples shown in *Figure 5*.

**Figure supplement 1.** Dual-colour DNA-PAINT sarcomere quantifications.

## A molecular map of the *Drosophila* titin homologs in flight muscle sarcomeres

Our data enabled us to build a molecular map of the *Drosophila* titin homologs in flight muscle sarcomeres, which revealed a significant overlap of the linear Sls and Projectin proteins at the I-band/A-band interface as visualised in a 'composite sarcomere' reconstructed by imaging flight muscles from six different hemi-thoraces (*Figure 6A*).

To precisely determine the position of all the epitopes investigated in our study, we calculated the average position using all the sarcomeres we imaged in the single- and dual-colour DNA-PAINT experiments. This strategy is valid as we found that although our mounting protocol for TIRF imaging results in a slightly variable sarcomere length of around 3.5 µm (*Spletter et al., 2015*), the distance between the measured epitopes is constant (*Figure 6—figure supplement 1*). Hence, the localisation of the different Sls and Projectin domains investigated using all sarcomeres measured resulted in a very high localisation precision with 95% confidence intervals of only 1–8 nm (*Figure 6B*). Pooling all data verified that the N-terminal Proj-Ig5-8 epitope is located 90 nm from the Z-disc, whereas the C-terminal Sls epitopes Sls-Ig49/50 and Sls-Ig51-Fn2 are located about 98 nm from the Z-disc. This is consistent with a staggered linear organisation of Sallimus and Projectin, which suggests an attractive mechanism how to mechanically link the sarcomeric Z-disc in insect flight muscle with the myosin filament using both titin homologs (*Figure 6C*).

## Discussion

### Super-resolution of flight muscles with nanobodies

The value of nanobodies and other small binders is well appreciated (*Harmansa and Affolter, 2018*). However, most *Drosophila* in vivo studies have thus far heavily relied on commercially available anti-GFP nanobodies to enhance GFP fluorescence signal in various tissues, including *Drosophila* flight muscles (*Kaya-Copur et al., 2021*) or to either trap GFP-fusion proteins ectopically or to degrade them when expressed in various modified forms in vivo (*Caussinus et al., 2011*; *Harmansa et al., 2015*; *Nagarkar-Jaiswal et al., 2015*). Our titin nanobody toolbox (*Loreau et al., 2023*) enabled us now to apply DNA-PAINT super-resolution technology to image the titin nanostructure in large intact flight muscle tissue at nanometre-scale resolution.

It had been shown that dye- or DNA-labelled nanobodies work well to achieve high labelling densities in cell culture (*Agasti et al., 2017*; *Fabricius et al., 2018*; *Mikhaylova et al., 2015*; *Pleiner et al., 2015*; *Schlichthaerle et al., 2019*). We have shown that our nanobodies are also very efficient in penetrating the large flight muscle fibres containing highly packed sarcomeres (*Loreau et al., 2023*), which are amongst the most protein-dense macromolecular structures in biology (*Daneshparvar et al., 2020*; *Taylor et al., 2019*). This high labelling efficiency enabled us to perform DNA-PAINT super-resolution microscopy of the large flight muscles without dissecting individual myofibrils. Such large specimens have rarely been investigated with DNA-PAINT (*Cheng et al., 2021*; *Lelek et al.,*

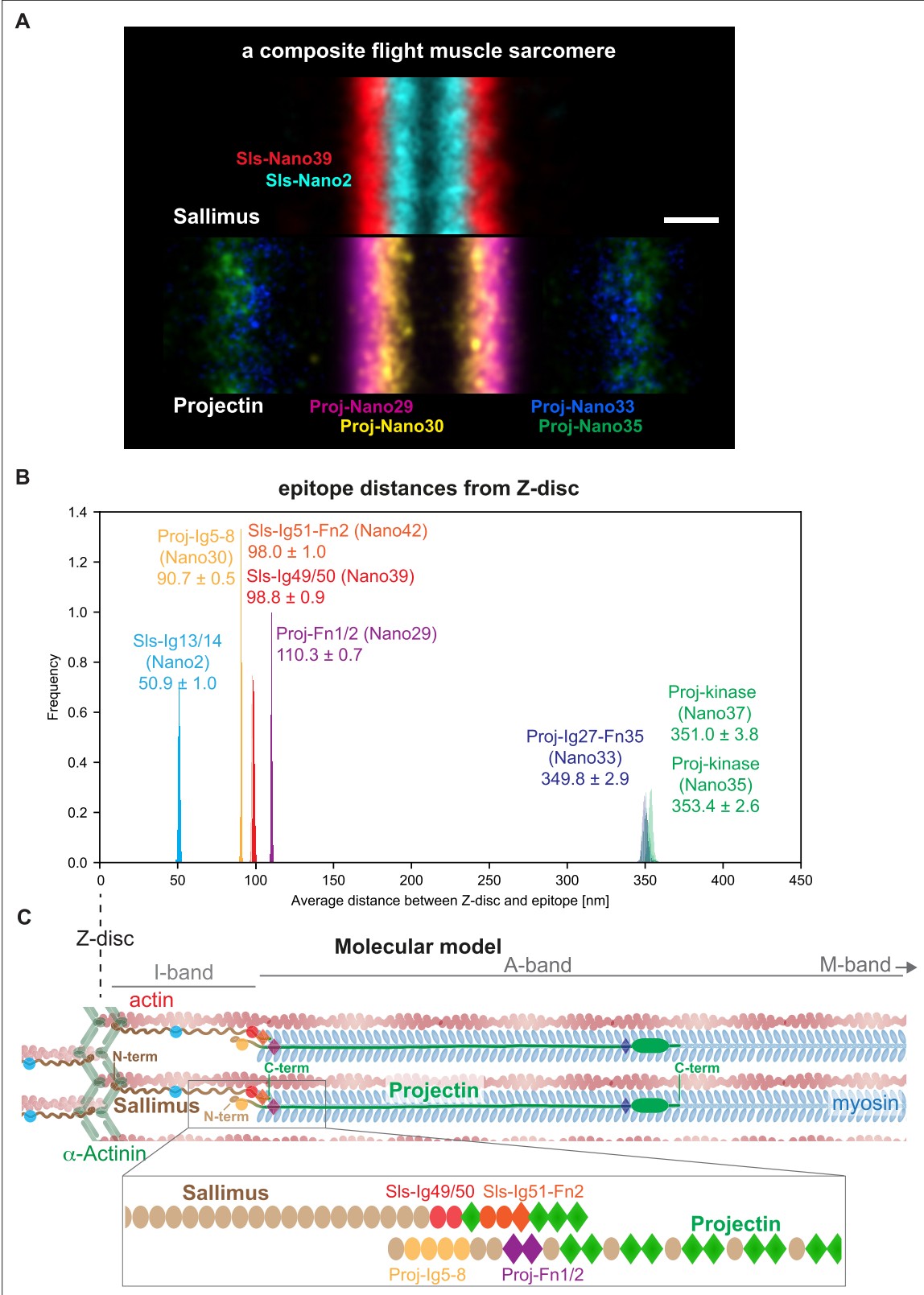

**Figure 6.** Summary and model. (**A**) A sarcomere displayed as a composite of two summed anti-Sallimus nanobody bands (top) and four summed anti-Projectin nanobody bands (bottom), each originating from one individual hemi-thorax imaged. Note the overlay of the positions of both proteins. Scale bar is 100 nm. (**B**) Distribution of the average distances from the Z-disc for all Sls and Projectin epitopes measured using bootstrapping (see 'Methods'). (**C**) Cartoon model of the relative arrangement of Sls and Projectin within the flight muscle sarcomere. The positions of the measured Sls and Projectin

*Figure 6 continued*

domains are highlighted in colours. The zoomed regions illustrate the suggested staggered architecture of the C-terminal Sls and the N-terminal Projectin protein parts.

The online version of this article includes the following source data and figure supplement(s) for figure 6:

**Source data 1.** Data summary and all band distances generated in the manuscript.

**Figure supplement 1.** Distance between bands versus sarcomere length.

*2021*). This shows that DNA-PAINT can be readily applied to super-resolve structures in large tissues if mounting and labelling protocols are optimised.

## Titin nanoarchitecture in flight muscles – Do titins rule?

Flight muscles are an ideal tissue to perform architectural studies of their sarcomeric components at the nanoscale because these components display an extremely high molecular order (*Loison et al., 2018*). This was impressively demonstrated by substructural averaging that resolved the nanostructure of myosin filaments isolated from insect flight muscles at a 7 Å resolution by cryo-electron-microscopy (*Daneshparvar et al., 2020*; *Hu et al., 2016*). Another recent study (*Szikora et al., 2020*) took advantage of this stereotypic order and used a series of existing antibodies against sarcomeric protein components to probe isolated myofibrils from *Drosophila* flight muscles using the super-resolution microscopy technique called STORM (*Rust et al., 2006*). The precisely reproducible sarcomeric morphology enabled averaging several hundred sarcomeres to reconstruct distances of various epitopes located at the Z-disc, including Zasp52 and α-Actinin, with 5–10 nm precision (*Szikora et al., 2020*). Although done on isolated dissected myofibrils and by averaging many sarcomeres, the large diversity of antibodies studied gave a comprehensive understanding of domain positions for a variety of important sarcomeric components. This included the Sls-Ig16 antibody, locating Sls-Ig16 about 50 nm from the centre of the Z-disc (*Szikora et al., 2020*), which is in good agreement with the location of Sls-Ig13/14 we found here. This study further showed that the Z-disc components α-Actinin and Zasp52 extend only about 35 nm from the centre of the Z-disc, whereas Filamin stretches with its C-term to about 55 nm (*Szikora et al., 2020*). This strongly suggests that the N-terminus of Sls, with its remaining 12 Ig domains, can reach and interact with these Z-disc components and possibly also with Filamin, as has been reported biochemically (*González-Morales et al., 2017*; *Liao et al., 2016*). Hence, the N-terminal part of the fly titin homolog Sls is arranged similarly to the N-terminus of vertebrate titin that binds to α-Actinin, anchoring it within the Z-disc (*Gautel and Djinović-Carugo, 2016*; *Ribeiro et al., 2014*). The exact molecular arrangement will need to await cryo-electron tomography data as recently achieved for the mammalian Z-disc (*Wang et al., 2021*).

An important part of the titin ruler model is that the titin spring part, which relaxes and stretches during muscle contraction and relaxation, respectively, spans across the I-band and sets the I-band length of vertebrate sarcomeres (*Brynnel et al., 2018*; *Linke, 2018*; *Luis and Schnorrer, 2021*). Thus, it is insightful that our newly developed C-terminal Sls nanobodies show that the C-terminal end of Sls is located about 100 nm from the centre of the Z-disc in flight muscles. This is consistent with the Sls-B2 antibody that supposedly recognises Sls-Ig44 (*Burkart et al., 2007*), which was localised to about 80 nm from the Z-disc (*Szikora et al., 2020*). However, this Sls-Ig44 is only present in a few Sls isoforms called 'Sls-700', which are likely expressed at early developmental stages at which the sarcomeres have just assembled. Hence, this antibody labels only the centre of each radially symmetric myofibril (*Burkart et al., 2007*). In contrast, our nanobodies bind Sls more C-terminally and show a wider band, suggesting that a larger number of Sls isoforms are reaching the myosin filaments. The fact that the C-terminal Sls nanobodies do not label the entire height of the myofibril (see *Figure 1B*) suggests that these epitopes are still not included in all the flight muscle Sls isoforms.

Although we have not imaged myosin directly in our samples, both STORM and electron-microscopy studies demonstrated that myosin filaments begin about 100 nm from the centre of the Z-disc, making the half I-band less than 100 nm wide (*Burkart et al., 2007*; *Kronert et al., 2018*; *Loison et al., 2018*; *Reedy and Beall, 1993*; *Szikora et al., 2020*). This strongly suggests that, as in vertebrates, Sls is indeed spanning across the short flight muscle I-band, where it could interact through its C-terminal domains with the myosin filament and hence could mechanically link the Z-disc with the myosin filament. Interestingly, the long isoform of the *C. elegans* titin homolog TTN-1 was shown to also bridge

across the I-band reaching the beginning of the A-band in *C. elegans* body muscles and in vitro studies demonstrated that its C-terminal Ig38-40 domains indeed bind to myosin with low nanomolar affinity (*Forbes et al., 2010*). These data are consistent with our hypothesis that *Drosophila* Sls may interact with myosin or Projectin and thus function as an I-band ruler in insect muscles (see model in *Figure 6C*).

The exact molecular arrangement of Sls in the I-band is still unknown. As the length of an isolated Ig domain is about 4 nm (*Mayans et al., 2001*), a chain of 40 Ig domains, which are present in the flight muscle Sls isoforms that we studied here, would result in a length of about 160 nm instead of the observed 100 nm. Hence, the Sls native structure in close proximity to actin filaments present in the flight muscle I-band might be more complex than a straight linear array of Ig domains.

The interpretation that Sls length rules I-band length is also supported by the observation that non-flight muscles like leg, jump, and larval muscles, which contain long I-bands, do express longer versions of Sls that include the large and flexible PEVK domains (*Burkart et al., 2007*; *Spletter et al., 2015*). Indeed, in the accompanying paper using the Sls nanobodies, we showed that Sls is more than 2 μm long in larval muscles to bridge over these long I-bands (*Loreau et al., 2023*). This strongly suggests that Sls determines I-band length in the different muscle types; however, a direct genetic test that modifies Sls length and assays I-band length remains to be done.

The vertebrate A-band contains the Ig-Fn super-repeats of titin, which extend from the beginning myosin filament until the M-band, where titin's C-terminal kinase is located (*Granzier et al., 2014*; *Lange et al., 2005*; *Linke, 2018*). Interestingly, we demonstrate that in *Drosophila* flight muscles, Projectin, which is very similar to the A-band part of vertebrate titin, with long Ig-Fn super-repeats and a C-terminal kinase domain, starts about 90 nm from the Z-disc. Hence, it is very unlikely that it can interact with Z-disc components directly as these are far from the N-terminal end of Projectin (model in *Figure 6C*). Our precise distance measurements suggest that the N-terminus of Projectin, which does contain a series of Ig domains, typical for the I-band part of titin, is sticking into the flight muscle I-band, whereas its first Fn/Ig super-repeat is located at beginning of the A-band (110 nm from the Z-disc) and hence can interact with myosin, as can its remaining Ig-Fn super-repeats that extend over a length of about 250 nm towards the M-band. This localisation differs somewhat from what was found by STORM of dissected myofibrils which placed the Ig domain 26 of Projectin only about 70 nm from the Z-disc (*Szikora et al., 2020*). We found that the Projectin kinase localises in a sharp band; however, it remains far from the M-band. Hence, it is hard to imagine that Projectin alone can directly rule the A-band length of flight muscle sarcomeres as it is only present at its distal ends, spanning about 15% of the myosin filament.

## Staggering insect titins to effectively transduce forces during flight?

*Drosophila* flight muscles are very stiff to effectively power wing oscillations during flight at 200 Hz. The perpendicular arrangement of the antagonistic dorsoventral (DVMs) versus the dorso-longitudinal flight muscles (DLMs) enables an effective stretch-activation mechanism as trigger: contraction of the DVMs moves the wings up and stretches the DLMs to induce their contraction, which will move the wings down again for the next cycle (*Dickinson et al., 2005*; *Pringle, 1981*; *Syme and Josephson, 2002*). The importance of strain in these muscles is highlighted by their expression of a particular troponin C isoform (TpnC4), which requires to be stretched to displace tropomyosin from myosin binding sites on actin filaments (*Agianian et al., 2004*). Furthermore, myosin also experiences a stretch-induced deformation before effective actin binding and maximum force production (*Iwamoto and Yagi, 2013*). This suggests that very effective force transmission is needed during flight muscle oscillations.

*Drosophila* sarcomeres contractions have a peak-to-peak amplitude of about 3.5% or 60 nm per half sarcomere during flight (measured in *Drosophila virilis*; *Chan and Dickinson, 1996*). This 3.5% strain is needed to produce the up to 110 W/kg power output of insect flight muscles (*Chan and Dickinson, 1996*), which is consistent with the hypothesis that strain across molecules stores the elastic energy for the next contraction cycle in *Drosophila* (*Dickinson et al., 2005*). A perfect candidate for such a molecule is Sls as it bridges across the I-band, which likely changes length during the fast contraction cycles. Thus, Sls length would oscillate during flight, which likely results in high oscillating forces across Sls during flight. A similar storage of elastic energy has been suggested for mammalian titin during sarcomere contraction cycles (*Eckels et al., 2019*; *Rivas-Pardo et al., 2020*).

What is the role of Projectin? The precise linear arrangement of Projectin at the beginning of the myosin filament and the found overlap with Sls suggests that this staggered architecture of Sls and Proj might be required to effectively anchor Sls to the myosin filament and to prevent sarcomere rupturing during flight. Such flight-induced muscle ruptures are generated when muscle attachment to tendons is weakened, underscoring the high muscle forces and high strain produced during flight (*Lemke et al., 2019*). Projectin may thus serve as an effective glue to stably connect Sls to the myosin filament. This is also consistent with the findings that both Sls and Projectin are needed to assemble contractile sarcomeres in *Drosophila* larval muscles. Knockdown of either protein results in embryonic lethality and defective sarcomerogenesis (*Loreau et al., 2023*; *Schnorrer et al., 2010*). Taken together, the staggered architecture of the two *Drosophila* titin homologs may effectively allow force transduction and ensure the mechanical integrity of flight muscles sarcomeres, both very prominent functions of mammalian titin (*Li et al., 2020*; *Rivas-Pardo et al., 2020*; *Swist et al., 2020*).

# Methods

**Key resources table**

| Reagent type (species) or resource | Designation | Source or reference | Identifiers | Additional information |
|---|---|---|---|---|
| Strain, strain background (*Drosophila melanogaster*) | Luminy | *Leonte et al., 2021* | | |
| Gene (*D. melanogaster*) | *sls* | http://flybase.org/reports/FBgn0086906 | FBgn0086906 | |
| Gene (*D. melanogaster*) | *bt* (Projectin) | http://flybase.org/reports/FBgn0005666 | FBgn0005666 | |
| Other | Sls-Ig13/14 (Nano2) | *Loreau et al., 2023* | | Nanobody – use at about 50 nM |
| Other | Sls-Ig49/50 (Nano39) | *Loreau et al., 2023* | | Nanobody – use at about 50 nM |
| Other | Sls-Ig51-Fn2 (Nano42) | *Loreau et al., 2023* | | Nanobody – use at about 50 nM |
| Other | Sls-Ig13/14 (Nano2) | *Loreau et al., 2023* | | Nanobody – use at about 50 nM |
| Other | Proj-Ig5-8 (Nano30) | *Loreau et al., 2023* | | Nanobody – use at about 50 nM |
| Other | Proj-Fn1/2 (Nano29) | *Loreau et al., 2023* | | Nanobody – use at about 50 nM |
| Other | Proj-Ig27-Fn35 (Nano33) | *Loreau et al., 2023* | | Nanobody – use at about 50 nM |
| Other | Proj-kinase (Nano37) | *Loreau et al., 2023* | | Nanobody – use at about 50 nM |
| Other | Proj-kinase (Nano35) | *Loreau et al., 2023* | | Nanobody – use at about 50 nM |
| Chemical compound, drug | Rhodamine-phalloidin | Invitrogen, Cat#R415 | | 1 in 500 |
| Chemical compound, drug | P1 imager – Atto643 | Metabion | TAGATGTAT – Atto643 | |
| Chemical compound, drug | P3 imager – Cy3b | Metabion | TAATGAAGA – Cy3B | |
| Chemical compound, drug | PS3 imager – Atto643 | Metabion | TCCTCCC – Atto643 | |
| Software, algorithm | PAINT data band extraction | https://github.com/PierreMangeol/titin_PAINT; *Mangeol, 2022* | | |

## Fly strains and fly culture

Fly stocks were grown and maintained under normal culture conditions in humidified incubators with 12 hr light–dark cycles on standard fly medium (*Avellaneda et al., 2021*). The particularly well-flying 'Luminy' strain was used in all experiments as wild type (*Leonte et al., 2021*). For all experiments, young 3–10-day-old flies were used.

## Nanobody production and labelling

Nanobody production and labelling with fluorophores by maleimide chemistry through ectopic cysteines was done as described in detail in the accompanying paper (*Loreau et al., 2023*). To couple nanobodies to DNA oligos, the oligos (P1, P2, PS3) were ordered with a 5' amino group modification (e.g., Am-C6-TTT CTT CAT TAC) from IBA (Göttingen) in HPLC-purified form and lyophilised as a triethylammonium (TEA) salt. Note that the absence of ammonia ($NH_4^+$) is essential for the procedure. 1 µmol of oligo was dissolved in 200 µl 30% acetonitrile (ACN), 15 mM TEA, which yielded a 5 mM stock at neutral pH (~7). 5 µl of a 100 mM cross-linker stock in 100% ACN (maleimido β-alanine NHS ester, Iris Biotec # MAA1020 or mal-PEG4-NHS, Iris Biotec # PEG1575) were added and allowed to react for 30 min on ice. Then, 1.6 µl 5 M sodium acetate, 0.1 M acetic acid (pH ~7) were added, and the modified oligo was precipitated by adding 1 ml 100% ACN and centrifugation for 10 min at 0°C at 12,000 rpm. This step removes any non-reacted maleimide. The pellet was then dissolved in 100 µl 30% ACN, and either stored in small aliquots at –80°C or used directly to label nanobodies at ectopic, reduced cysteines, as described for fluorophores in the accompanying paper (*Loreau et al., 2023*). Note that free oligo cannot be removed by gel filtration on Sephadex G25 because it appears with the conjugate in the void volume. It is best removed by modifying a still His14-SUMO or His14-NEDD8 tagged nanobody (*Frey and Görlich, 2014*; *Pleiner et al., 2015*) and then using Ni(II) capture (where the free oligo remains non-bound) and proteolytic release of the then tag-free nanobody conjugate. The efficiency of conjugation can be assessed by SDS-PAGE, in which the oligo-modification results in a clear size shift. In addition, the density of modification can be calculated through OD260 and OD280 readings using ε260 and ε280 of the initial oligo and nanobody as input variables. The oligo modification by this method is usually quantitative already with a small (≥1.1) molar excess of the maleimide oligo over modifiable cysteines. In case of incomplete modification, the conjugate can be purified on a MonoQ column, whereby the highly negative charged oligo causes stronger retention of the conjugate compared to the non-modified nanobody.

## Flight muscle preparation, staining, and mounting for imaging

Intact hemi-thoraces from adult males were prepared similar as described (*Weitkunat and Schnorrer, 2014*). Head, wings, and abdomen were clipped with sharp forceps and the intact thoraces were fixed for 20 min at room temperature in relaxing solution (4% PFA in 100 mM NaCl, 20 mM NaPi pH 7.2, 6 mM $MgCl_2$, 5 mM ATP, 0.5% Triton X-100). After washing twice with relaxing solution, the thoraces were placed on a slide with double-sticky tape and cut sagittally with a sharp microtome blade (Pfm Medical Feather C35). The fixed hemi-thoraces were transferred to 24-well plates or Eppendorf tubes and blocked for 30 min at room temperature with 3% normal goat serum in PBS + 0.5% Tx-100 (PBS-T). Hemi-thoraces were stained overnight at 4°C with the combinations of nanobodies indicated, labelled with fluorophores or oligonucleotides (final concentration of about 50 nM). Actin was stained with phalloidin-rhodamine or phalloidin-Alexa488 (1:2000, Thermo Fisher; 2 hr at room temperature or overnight at 4°C). To mount the flight muscles as close as possible to the coverslip, an imaging chamber was built using a slide and #1 coverslips as spacers right and left of the samples. A layer of double sticky tape was built on the spacer and the imaging chamber was filled with either Slow-Fade Gold Antifade (Thermo Fisher) for confocal imaging or Imager solution for DNA-PAINT imaging. Stained hemi-thoraces were added, oriented with the flight muscles facing up and #1.5 coverslip was added. The chamber was sealed with nail polish for confocal imaging or Picodent glue for DNA-PAINT imaging.

## Confocal imaging

Stained flight muscles were imaged on a Zeiss LSM880 confocal with a ×63 oil lens. Images were processed using Fiji (*Schindelin et al., 2012*).

## DNA-PAINT imaging

### Materials

Cy3B-modified and Atto643-modified DNA oligonucleotides were custom-ordered from Metabion. Sodium chloride 5 M (cat#: AM9759) was obtained from Ambion. Coverslips (cat#: 0107032) and glass slides (cat#: 10756991) were purchased from Marienfeld and Thermo Fisher. Double-sided tape (cat#: 665D) was ordered from Scotch. Two-component silica twinsil speed 22 (cat#: 1300 1002)

was ordered from picodent. Glycerol (cat#: 65516-500ml), methanol (cat#: 32213-2.5L), protocate-chuate 3,4-dioxygenase pseudomonas (PCD) (cat#: P8279), 3,4-dihydroxybenzoic acid (PCA) (cat#: 37580-25G-F) and (+−)–6-hydroxy-2,5,7,8- tetra-methylchromane-2-carboxylic acid (Trolox) (cat#: 238813-5G) were ordered from Sigma. Potassium chloride (cat#: 6781.1) was ordered from Carl Roth. Paraformaldehyde (cat#: 15710) was obtained from Electron Microscopy Sciences. 90 nm diameter Gold Nanoparticles (cat#: G-90-100) were ordered from Cytodiagnostics.

## Buffers

For imaging, the following buffer was prepared: Buffer C (1× PBS, 500 mM NaCl). Directly before imaging Buffer C was supplemented with 1× Trolox, 1× PCA and 1× PCD (see paragraph below for details). 100× Trolox: 100 mg Trolox, 430 µl 100% methanol, 345 µl 1 M NaOH in 3.2 ml $H_2O$. 40× PCA: 154 mg PCA, 10 ml water and NaOH were mixed, and pH was adjusted to 9.0. 100× PCD: 9.3 mg PCD, 13.3 ml of buffer (100 mM Tris-HCl pH 8, 50 mM KCl, 1 mM EDTA, 50% glycerol). All three were frozen and stored at –20°C.

## Sample preparation

*Drosophila* hemi-thoraces were isolated and stained as described above with phalloidin Alexa488 (1:2000) and the two nanobodies coupled to either P1, P3, or PS3 oligos (about 50 nM) overnight. Before embedding the samples into the chamber, they were washed two times with PBS + 1% Triton. Hemi-thoraces were embedded as described above. Before assembling the chamber, the cover slip was treated with 90 nm diameter gold nanoparticles (cat#: G-90-100, Cytodiagnostics, 1:10 dilution into methanol). After assembling, the chamber was filled with imaging buffer containing the complementary P1, P3, or PS3 imaging oligos (see below for imaging conditions) and sealed with Picodent glue.

## Super-resolution microscope

Fluorescence imaging was carried out on an inverted microscope (Nikon Instruments, Eclipse Ti2) with the Perfect Focus System, applying an objective-type TIRF configuration with an oil-immersion objective (Nikon Instruments, Apo SR TIRF 100×, NA 1.49, Oil). A 561 nm and 640 nm (MPB Communications Inc, 2W, DPSS-system) laser were used for excitation. The laser beam was passed through clean-up filters (Chroma Technology, ZET561/10, ZET642/20x) and coupled into the microscope objective using a beam splitter (Chroma Technology, ZT561rdc, ZT647rdc). Fluorescence light was spectrally filtered with an emission filter (Chroma Technology, ET600/50m and ET575lp, ET705/72m and ET665lp) and imaged on a sCMOS camera (Andor, Zyla 4.2 Plus) without further magnification, resulting in an effective pixel size of 130 nm (after 2 × 2 binning).

## Imaging conditions

See *Figure 4—source data 1* and *Figure 5—source data 1*.

## **Imager sequences**

> P1-Atto643; Imager: TAGATGTAT – Atto643; Nanobody – TTATACATCTA;
> P3-Cy3b; Imager: TAATGAAGA – Cy3B; Nanobody – TTTCTTCATTA;
> PS3-Atto643; Imager: TCCTCCC – Atto643; Nanobody – AAGGGAGGA.

## Super-resolved image reconstruction

The data acquired during imaging was post-processed using the Picasso (*Schnitzbauer et al., 2017*) pipeline. First, the localisations were detected by a threshold-based detection and fitted with a least-square fit; the resulting localisation precision was estimated between 4 and 8 nm using the NeNA metric (*Endesfelder et al., 2014*). Next, the data was drift-corrected using a redundant cross-correlation and a fiducial marker-based drift correction. Then, a super-resolved image was rendered using Picasso render. From the images, the myofibrils for further analysis were selected interactively using the rectangular pick tool. All further analysis was done with customised Jupyter Notebooks.

## Extraction of band positions from DNA-PAINT data

Extraction of band positions from DNA-PAINT data was achieved the following way: first, individual myofibrils were manually selected using the rectangular selection tool from Picasso (*Schnitzbauer et al., 2017*) and saved in individual files.

Second, the remaining of the analysis was automated in custom codes written in Python. To limit localisation events arising from multiple emitters that create artefacts (*Lelek et al., 2021*), localisations were filtered based on the standard deviation of their Gaussian fits. Localisations kept were within a disc in the standard deviation space (sx, sy), centred on the maximum of the distribution and of radius 0.2 pixel.

Third, individual Z-discs were automatically detected. This did not require super-resolved data and the process was the result of multiple steps: (a) the algorithm rotated selections and their localisations to orient the selection horizontally. (b) Localisations were projected along the main axis of the selection and their density was reported in a histogram, in which bin size was the same as the pixel size of the camera. The histogram can be seen as a low-resolution intensity profile along the myofibril. (c) The algorithm found peaks in the resulting histogram (with find_peaks from the SciPy library) corresponding to positions of Z-discs. (d) Once peaks were detected, the algorithm selected peaks that were relevant to the analysis using the fact that the distance between Z-discs is the size of a sarcomere.

Fourth, with the knowledge of Z-disc positions, the algorithm then focused on windows centred on Z-discs to extract the positions of bands: (a) Similar to step 3, the algorithm rotated the selection and stored localisations in a histogram, in which bin size is adjusted for best results (typical bin size was 13 nm). (b) Because DNA-PAINT data accumulate the localisations, the histogram of localisations can display fluctuations that make automated extraction of band positions difficult. Therefore, to locate the rough position of a given band, the data were first convolved with a Gaussian function of standard deviation 25 nm that smoothens fluctuations. (c) The resulting histogram was then analysed with a peak-finding algorithm to locate rough band positions. (d) Finally, to precisely locate band positions, the algorithm fitted a Gaussian function on the non-convolved data, in a window centred on each of the positions detected at the previous step. To ensure that the analysis was properly achieved, the results were visually checked. The code is available at https://github.com/PierreMangeol/titin_PAINT; *Mangeol, 2022*.

## Average epitope positions using bootstrapping

To obtain an uncertainty estimate of the average position of epitopes, we used the bootstrapping method (*Efron and Tibshirani, 1994*). In brief, each dataset of an epitope is used to create 1000 bootstrap replicates. We generated a replicate by drawing individual values in a given dataset with replacement (i.e. each value can be drawn multiple times). The size of one replicate is the same as the one of the initial dataset. From each of these replicates, we computed the mean, and therefore obtained 1000 means from 1000 replicates. These 1000 means constitute the bootstrap data presented in *Figure 6*, each epitope having its own bootstrap data. Finally, 95% confidence intervals were obtained by extracting the 2.5% and 97.5% quantiles from these bootstrap data.

## Materials availability statement

Newly generated code is publicly available here at https://github.com/PierreMangeol/titin_PAINT, (copy archived at swh:1:rev:95e2ac29f658f8fca2435d93ab3c6326c786047d; *Mangeol, 2022*).

Nanobodies are described in *Loreau et al., 2023*, and expression plasmids will be made available from Addgene.

## Acknowledgements

We thank Sandra B Lemke and Aynur Kaya-Çopur for their help in the initial DNA-PAINT pilot experiments. We would like to thank Stefan Raunser and Mathias Gautel and all their group members, as well as the Schnorrer and Görlich groups, for their stimulating discussions within the StuDySARCO-MERE ERC synergy grant. We are indebted to the IBDM imaging facility for help with image acquisition and maintenance of the microscopes. This work was supported by the Centre National de la Recherche Scientifique (CNRS, F Schn), the Max Planck Society (RJ, DG), Aix-Marseille University

(PM), the European Research Council under the European Union's Horizon 2020 Programme (ERC-2019-SyG 856118 to DG and F Schn and ERC-2015-StG 680241 to RJ), the German Research Foundation through the SFB1032 (Project-ID 201269156 to RJ), the excellence initiative Aix-Marseille University A*MIDEX (ANR-11-IDEX-0001-02, FS), the French National Research Agency with ANR-ACHN MUSCLE-FORCES (FS), the Human Frontier Science Program (HFSP, RGP0052/2018, FS), the Bettencourt Foundation (FS), the France-BioImaging national research infrastructure (ANR-10-INBS-04-01), and by funding from France 2030, the French Government program managed by the French National Research Agency (ANR-16-CONV-0001) and from Excellence Initiative of Aix-Marseille University-A*MIDEX (Turing Centre for Living Systems). The funders had no role in study design, data collection and analysis, decision to publish, or preparation of the manuscript.

## Additional information

### Funding

| Funder | Grant reference number | Author |
|---|---|---|
| Centre National de la Recherche Scientifique | | Frank Schnorrer |
| Aix-Marseille Université | | Pierre Mangeol |
| Max-Planck-Gesellschaft | | Dirk Görlich<br>Ralf Jungmann |
| European Research Council | ERC-2019-SyG 856118 | Dirk Görlich<br>Frank Schnorrer |
| European Research Council | ERC-2015-StG 680241 | Ralf Jungmann |
| Aix-Marseille Université | ANR-11-IDEX-0001-02 | Frank Schnorrer |
| Agence Nationale de la Recherche | ACHN MUSCLE-FORCES | Frank Schnorrer |
| Human Frontier Science Program | RGP0052/2018 | Frank Schnorrer |
| Agence Nationale de la Recherche | ANR-10-INBS-04-01 | Frank Schnorrer |
| Agence Nationale de la Recherche | ANR-16-CONV-0001 Turing Centre for Living Systems | Frank Schnorrer |

The funders had no role in study design, data collection and interpretation, or the decision to submit the work for publication. Open access funding provided by Max Planck Society.

### Author contributions

Florian Schueder, Pierre Mangeol, Data curation, Formal analysis, Investigation, Visualization, Methodology, Writing - original draft, Writing - review and editing; Eunice HoYee Chan, Data curation, Formal analysis, Investigation, Writing - review and editing; Renate Rees, Jürgen Schünemann, Data curation, Investigation, Methodology; Ralf Jungmann, Conceptualization, Supervision, Funding acquisition, Methodology, Writing - review and editing; Dirk Görlich, Conceptualization, Formal analysis, Supervision, Funding acquisition, Investigation, Methodology, Writing - original draft, Writing - review and editing; Frank Schnorrer, Conceptualization, Data curation, Formal analysis, Supervision, Funding acquisition, Investigation, Visualization, Methodology, Writing - original draft, Writing - review and editing

### Author ORCIDs

Pierre Mangeol http://orcid.org/0000-0002-8305-7322
Eunice HoYee Chan http://orcid.org/0000-0003-3162-3609
Ralf Jungmann http://orcid.org/0000-0003-4607-3312
Dirk Görlich http://orcid.org/0000-0002-4343-5210

Frank Schnorrer  http://orcid.org/0000-0002-9518-7263

**Decision letter and Author response**
Decision letter https://doi.org/10.7554/eLife.79344.sa1
Author response https://doi.org/10.7554/eLife.79344.sa2

---

## Additional files

### Supplementary files
• MDAR checklist

### Data availability
All source data for all figures are provided. Figure 3 - Source Data 1 Figure 4 - Source Data 1 Figure 5 - Source Data 1 Figure 6 - Source Data 1. All new code has been uploaded to a public database and can be found at https://github.com/PierreMangeol/titin_PAINT, copy archived at swh:1:rev:95e2ac29f658f8fca2435d93ab3c6326c786047d.

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
