## [Editor Report]

This landmark study combines two advanced technologies, namely, nanobodies and DNA-PAINT, to define the position of several subdomains of the two fly titin homologs in adult flight muscles. Their results provide compelling evidence that Sallimus can bridge the Z-disk with the beginning of the myosin A-band. Furthermore, their results convincingly establish that Sallimus and Projectin partially overlap at the beginning of the A-band. The work should appeal to people interested in muscle biology and more generally to people interested in providing high-resolution images of long proteins.

---

## [Decision Letter]

**Decision letter after peer review:**

Thank you for submitting your article "Nanobodies combined with DNA-PAINT super-resolution reveal a staggered titin nano-architecture in flight muscles" for consideration by *eLife*. Your article has been reviewed by 4 peer reviewers, and the evaluation has been overseen by a Reviewing Editor and Anna Akhmanova as the Senior Editor. The following individuals involved in review of your submission have agreed to reveal their identity: Michel Labouesse (Reviewer #1); Guy M Benian (Reviewer #3).

Essential revisions:

The three external reviewers for each of your two manuscripts, along with myself as a guest editor, are overall supportive of your work. We think that you are introducing interesting tools, although one of the reviewers thinks that you overemphasized the novelty of nanobodies as an approach. The model you propose for sarcomere localization of the titin-like fly homologs Sallimus and Projectin is interesting and should inspire further work.

That being said, after exchanging views among the reviewers, we feel that there is probably too much redundancy between the two manuscripts in terms of methods and overall conclusions.

1) We suggest that you merge both manuscripts. This would allow you to show that your data are consistent among the two different muscle types (leg muscle and flight muscle), with an overlap in localization of the C-term of Sls with the N-term of Projectin at the outer edge of the A-band. Generation of the nanobodies and their ability to penetrate tissues would be described only once.

2) In terms of additional experiments, we ask that you use a nanobody-GFP to determine whether expression of the nanobody fused to mNeonGreen in muscle might affect muscle function or affect behavior of the fusion. You could easily test this by performing some sort of motility assays on the larva and determine whether muscle sarcomere structure is affected after immunostaining with various marker antibodies. Alternatively, you could use the nanobody-GFP to follow the course of a contraction/relaxation cycle in vivo. When reorganizing your manuscripts into a single one, please give better credit to the *C. elegans* muscle field (as requested by one reviewer), and generally attend to other minor comments formulated by the reviewers.

*Reviewer #1 (Recommendations for the authors):*

My main reservation over the two papers by Schueder et al. and by Loreau et al. concerns their level of redundancy. The Schueder et al. work is certainly more innovative and goes one step further in terms of biological conclusions than its companion manuscript by Loreau et al. In particular it makes an interesting comparison with vertebrate titin.

As it stands, Figures 1, 4, 5 of the Loreau et al. ms and much of Figures 1, 2 in this one are the same; furthermore, the Loreau et al. ms looks at the organization of different Sallimus isoforms in different muscle types, whereas this one looks at short isoforms in flight muscles, but the objectives are similar. One avenue to make the two manuscripts more clearly different would be to keep the presentation of nanobody isolation and penetration qualities to a single manuscript. A parallel strategy is to push the characterization of the use of nanobodies in vivo in the other manuscript and to explore whether the Sallimus C-ter and Projectin N-ter interact over their staggered overlapping area as the authors suggest at the end of their discussion, or whether the Sallimus C-ter could interact with myosin as the authors also raise in their discussion.

*Reviewer #2 (Recommendations for the authors):*

The manuscript is very clear and neatly presented. I have very few comments on the experimental design or the results, which follow very high standards. An apparent limitation to the study would reside in the lack of functional analysis to explore the insight gained from this work, and the functional role of the overlap between the two titin homologs. The study however is self sufficient, presents interesting and significant results, and describes a very clear experimental design which might be helpful to others in the field attempting to address similar questions.

Specific comments:

Lines 200-219: The DNA-PAINT description of chemical coupling is detailed but a figure would be helpful to support the text, showing reactions with structural formula.

Lines 515-516, Figure 1, Figure 2: A more detailed description of classical confocal microscopy setup (including pixel size) would be useful.

Line 574: I could not find the number of independent samples/replicates (number of experimental replicates, individuals, muscles, sarcomeres) supporting the author's experimental results. In addition to the sample ID used for the microscopy images that are presented, it would thus be interesting to provide a statistics table including this information and in particular showing, for key quantitative results, the number of samples from which the data are derived.

Line 580: An explicit description of the localisation precision for individual particles in the considered imaging conditions would be useful.

*Reviewer #3 (Recommendations for the authors):*

Overall, this is a very nicely executed and described study! I have only a few minor suggestions for interpretation and mostly on writing to improve clarity:

1. lines 115-117: In referring to exons in the gene, please change, "C-terminally located exons" to "3' located" or "3'-most located" exons.

2. lines 148-152: "These data demonsrate that Projectin is present in an extended conformation and since the flight muscle I-band extends less than 100 nm from the Z-disc…, a large part of Project is present along the myosin filament." Are there reports of EM imaging of isolated Projectin with contour length measurements? Given the domain composition and molecular mass, it is likely that Projectin is about 200 nm long. Can this be taken into account in their description?

3. lines 263-265: "The N-terminus of Sls is located close to the Z-disk, with Ig13/14 only about 50 nm away from the center of the Z-disc…" Can the authors be more definitive in their description, perhaps saying something like, "The result that nanobodies to Ig13/14 localize 50 nm away from the center of the Z-disc, is compatible with the very N-terminus of Sls being located at the center of the Z-disc, because the more N-terminal 12 Ig domains are likely to span 48 nm (longest dimension of Ig domain = 4 nm X 12 = 48 nm).

4. lines 282-284 and Figure 5A: "Our results verifeid that Sls-Ig13/14 is localizised about 50 nm away from the center of the Z-disk and that Sls-Ig51/Fn2 is about 50 nm further towards the middle of the sarcomere." Given that in the Sls domain map, Ig13/14 and Ig51/Fn2 are about 27 Ig domain from each other, wouldn't the authors expect that the observed distance should be about 108 nm? (27 Ig X 4 nm = 108 nm). And if so, can the authors say here that this implies that in the flight muscle imaged, this portion of Sls is not in an extended but likely "jumbled" or "compressed" chain of Ig domains?

5. For all the images presented, how do you know where the center of the Z-disk is located? There do not appear to be any co-stainings with antibodies or nanobodies to α-actinin.

6. line 385: Given that the I-band is bisected by the I-band, shouldn't the authors say here "half I-band" rather than "I-band"?

7. In Figure 1B, it seems like three of the nanobodies (39, 42 and 48) localize to Z-disks but not to the bottom (or even top) of the Z-disks compared to nanobody 2, which seems to localize the full height of the Z-disk. Can the authors comment on this?

8. Figure 6 is nicely presented. However, to help the reader, in part B, instead of listing the nanobody number, could the authors indicate to which domains the nanobodies were made? In part C, again to help the readers understand the full impact of the excellent results, could the authors indicate the N- and C-termini of Sls and Projectin?

*Reviewer #4 (Recommendations for the authors):*

Overall, as it stands, this manuscript even if of high technological value, remains entirely descriptive and short in providing new insights into muscle structure and architecture. The main finding, an overlap between short Sls isoform and Proj in flight muscle sarcomeres, is redundant with the author's observation in the companion paper that in larval muscles expressing a long Sls isoform, Sls and Proj overlap as well.

I would not recommend this manuscript for publication in *eLife*. It appears more suited for a specialized methodological development journal.

Combination of Sls and Proj nanobodies with DNA-Paint represents an attractive example of technological development which alternatively could strengthen the co-submitted nanobodies toolkit manuscript.

---

## [Author Response]

Essential revisions:The three external reviewers for each of your two manuscripts, along with myself as a guest editor, are overall supportive of your work. We think that you are introducing interesting tools, although one of the reviewers thinks that you overemphasized the novelty of nanobodies as an approach. The model you propose for sarcomere localization of the titin-like fly homologs Sallimus and Projectin is interesting and should inspire further work.

We are grateful to the reviewers, especially for their efforts to improve our manuscripts. We are happy that the reviewers share the interest in the tools generated here, in the DNA-PAINT method as well as in the biological findings resulting from the DNA-PAINT measurements in flight muscles.

That being said, after exchanging views among the reviewers, we feel that there is probably too much redundancy between the two manuscripts in terms of methods and overall conclusions.

As explained in our initial rebuttal letter, we have the opinion that both papers should stand alone. To substantiate this, we have largely expanded the nanobody toolbox paper. We provide all the details of this expansion in the response to reviewers’ letter of the toolbox paper. Briefly, we have added 10 additional new nanobodies against different domains of Obscurin, α-Actinin and Zasp52 to the resource paper to make the resource more complete.

We have further included the asked characterisations of the larvae expressing the NeonGreen tagged nanobodies and generated additional Nanobody-NeonGreen tagged fly strains. Finally, we provide a first analysis of nanobodies tagged with a deGrad signal in vivo. Furthermore, to remove redundancy, we have moved the nanobody penetration data in flight muscles from the PAINT paper to the toolbox paper and also moved almost all the confocal analysis to the toolbox paper, as requested. We will be ready to submit the revised toolbox paper in a few days, too.

1) We suggest that you merge both manuscripts. This would allow you to show that your data are consistent among the two different muscle types (leg muscle and flight muscle), with an overlap in localization of the C-term of Sls with the N-term of Projectin at the outer edge of the A-band. Generation of the nanobodies and their ability to penetrate tissues would be described only once.

We have moved almost all the confocal microscopy part including the quantitative antibody penetration analysis to the toolbox manuscript. We only left a basic Figure1 in the DNA-PAINT manuscript to be able to understand it for the reader as a stand-alone paper (see also below in the reviewers’ comments why this is important). Hence, the DNA-PAINT manuscript does now only include flight muscle data.

We want to point out that one major finding is that Projectin organisation is different in flight and larval muscle. In flight muscles, Projectin is only located at the tip of the myosin filament, overlapping the end of Sallimus (see PAINT paper), whereas in the larval muscle Projectin decorates the entire thick filament, likely in an oriented fashion (see Loreau et al). The latter assembly still allows a possible overlap of Sallimus and Projectin at the myosin filament ends, but this overlap is still hypothetical to date. Importantly, only a fraction of Projectin molecules will be able to do so. Hence, both are independent biological findings and only the flight muscle part was thus far analysed with super-resolution.

(2) In terms of additional experiments, we ask that you use a nanobody-GFP to determine whether expression of the nanobody fused to mNeonGreen in muscle might affect muscle function or affect behavior of the fusion. You could easily test this by performing some sort of motility assays on the larva and determine whether muscle sarcomere structure is affected after immunostaining with various marker antibodies. Alternatively, you could use the nanobody-GFP to follow the course of a contraction/relaxation cycle in vivo. When reorganizing your manuscripts into a single one, please give better credit to the *C. elegans* muscle field (as requested by one reviewer), and generally attend to other minor comments formulated by the reviewers.

Thanks for these valuable suggestions, we have done these experiments and text changes. We detail these changes in the response letter to this manuscript.

Reviewer #1 (Recommendations for the authors):My main reservation over the two papers by Schueder et al. and by Loreau et al. concerns their level of redundancy. The Schueder et al. work is certainly more innovative and goes one step further in terms of biological conclusions than its companion manuscript by Loreau et al. In particular it makes an interesting comparison with vertebrate titin.As it stands, Figures1, 4, 5 of the Loreau et al. ms and much of Figures 1, 2 in this one are the same; furthermore, the Loreau et al. ms looks at the organization of different Sallimus isoforms in different muscle types, whereas this one looks at short isoforms in flight muscles, but the objectives are similar. One avenue to make the two manuscripts more clearly different would be to keep the presentation of nanobody isolation and penetration qualities to a single manuscript. A parallel strategy is to push the characterization of the use of nanobodies in vivo in the other manuscript and to explore whether the Sallimus C-ter and Projectin N-ter interact over their staggered overlapping area as the authors suggest at the end of their discussion, or whether the Sallimus C-ter could interact with myosin as the authors also raise in their discussion.

We thank the reviewer for his valuable suggestions. We have entirely removed old Figure 2 (nanobody penetration in flight muscles) and replaced it with a new Figure 2 showing the chemistry of the oligo coupling following the request of reviewer 2. We have also strongly slimmed Figure 1, now only showing the dominant flight muscle splice isoforms of Sls and Projectin in the supplement and one representative set of nanobodies analysed with confocal resolution to justify the following super resolution experiments. We feel that without this basic figure the paper is less well understandable on its own (see point 2 and reviewers 2 and 3). We hope that this reviewer will agree with our solution.

Reviewer #2 (Recommendations for the authors):The manuscript is very clear and neatly presented. I have very few comments on the experimental design or the results, which follow very high standards. An apparent limitation to the study would reside in the lack of functional analysis to explore the insight gained from this work, and the functional role of the overlap between the two titin homologs. The study however is self sufficient, presents interesting and significant results, and describes a very clear experimental design which might be helpful to others in the field attempting to address similar questions.

We thank this reviewer for her/his enthusiastic evaluation of the quality of our results and their significance.

Specific comments:Lines 200-219: The DNA-PAINT description of chemical coupling is detailed but a figure would be helpful to support the text, showing reactions with structural formula.

We followed the reviewer’s advice and have added a new Figure 2 detailing the chemistry that was applied to couple the DNA oligo to the nanobody. Thanks for pointing this out.

Lines 515-516, Figure 1, Figure 2: A more detailed description of classical confocal microscopy setup (including pixel size) would be useful.

Following the general advice from the editor we have removed old Figure 2 and reduced Figure 1. Pixel size on a scanning confocal is somewhat arbitrarily set, as one can digitally zoom and set the resolution at the microscope resulting in a strong oversampling of the image beyond the technical resolution of the microscope. We have added a brief explanation to the text that the typical resolution of the confocal microscopy technique used here is limited to about 250 nm. This point again shows why we believe it is necessary to keep a slimmed Figure 1.

Line 574: I could not find the number of independent samples/replicates (number of experimental replicates, individuals, muscles, sarcomeres) supporting the author's experimental results. In addition to the sample ID used for the microscopy images that are presented, it would thus be interesting to provide a statistics table including this information and in particular showing, for key quantitative results, the number of samples from which the data are derived.

These data were in the Source data table. We have now provided Source data tables for the Figure 3, 4, 5 and 6 separately and added a summary providing numbers for animals, sarcomeres and Z-discs analysed. So, all the data are within the manuscript. The number of sarcomeres measured in the samples shown in the figures can also be seen in the charts on the right side of the figure in which we show ‘frequency’, which is the number of sarcomeres in a particular length bin shown. We added a sentence to the figure legends to make this clearer. We also added the total amount of sarcomeres scored in each sample into the figure legends of Figure 4 and Figure 5.

In Figure 6 supplement 1 each dot is a distance measure from one nanobody per sarcomere. A total of 1442 distances were measured. We added a sentence to the figure legends to make this clearer.

Line 580: An explicit description of the localisation precision for individual particles in the considered imaging conditions would be useful.

We have added the NeNA metric method that we used to estimate the resolution to the Methods section. Thanks for pointing this out.

Reviewer #3 (Recommendations for the authors):Overall, this is a very nicely executed and described study! I have only a few minor suggestions for interpretation and mostly on writing to improve clarity:1. lines 115-117: In referring to exons in the gene, please change, "C-terminally located exons" to "3' located" or "3'-most located" exons.

Done, thanks.

2. lines 148-152: "These data demonsrate that Projectin is present in an extended conformation and since the flight muscle I-band extends less than 100 nm from the Z-disc…, a large part of Project is present along the myosin filament." Are there reports of EM imaging of isolated Projectin with contour length measurements? Given the domain composition and molecular mass, it is likely that Projectin is about 200 nm long. Can this be taken into account in their description?

We are not aware of any studies showing full-length Projectin single molecule analysis with EM. We find a Projectin length of 250 nm, so close to the theoretical estimation of this reviewer (there are some linker sequences between the domains, that are omitted in the domains scheme).

3. lines 263-265: "The N-terminus of Sls is located close to the Z-disk, with Ig13/14 only about 50 nm away from the center of the Z-disc…" Can the authors be more definitive in their description, perhaps saying something like, "The result that nanobodies to Ig13/14 localize 50 nm away from the center of the Z-disc, is compatible with the very N-terminus of Sls being located at the center of the Z-disc, because the more N-terminal 12 Ig domains are likely to span 48 nm (longest dimension of Ig domain = 4 nm X 12 = 48 nm).

Thanks for this very accurate suggestion!

4. lines 282-284 and Figure 5A: "Our results verifeid that Sls-Ig13/14 is localizised about 50 nm away from the center of the Z-disk and that Sls-Ig51/Fn2 is about 50 nm further towards the middle of the sarcomere." Given that in the Sls domain map, Ig13/14 and Ig51/Fn2 are about 27 Ig domain from each other, wouldn't the authors expect that the observed distance should be about 108 nm? (27 Ig X 4 nm = 108 nm). And if so, can the authors say here that this implies that in the flight muscle imaged, this portion of Sls is not in an extended but likely "jumbled" or "compressed" chain of Ig domains?

We have modified our discussion to address this interesting point, to which we do not have a definitive answer.

5. For all the images presented, how do you know where the center of the Z-disk is located? There do not appear to be any co-stainings with antibodies or nanobodies to α-actinin.

We have imaged phalloidin to locate the samples as described in the methods. We do not display this information as we have not imaged phalloidin at super resolution. However, we know that the 2 bands around the Z-disc are always symmetric, see Figure 1 or see Loreau et al. 2022. So this is not a concern.

6. line 385: Given that the I-band is bisected by the I-band, shouldn't the authors say here "half I-band" rather than "I-band"?

We assume the reviewer means, ‘bisected by the Z-disc’, yes this is a good point. As classical sarcomere descriptions define the I-band as continuous, half I-band is more accurate. We changed it in the manuscript.

7. In Figure 1B, it seems like three of the nanobodies (39, 42 and 48) localize to Z-disks but not to the bottom (or even top) of the Z-disks compared to nanobody 2, which seems to localize the full height of the Z-disk. Can the authors comment on this?

Very well spotted by this reviewer. This is true and the likely reason is that not all the long Sls isoforms do contain these epitopes. We have added a comment in the discussion to it, at the point where we do discuss the old Sls antibody “B2” that only labels the very central region of each myofibril and is against a more N-terminal domain that is present in even fewer isoforms.

8. Figure 6 is nicely presented. However, to help the reader, in part B, instead of listing the nanobody number, could the authors indicate to which domains the nanobodies were made? In part C, again to help the readers understand the full impact of the excellent results, could the authors indicate the N- and C-termini of Sls and Projectin?

As we have different nanobodies from some of the domains, we prefer to keep the nanobody numbers. But we added the domains as well to make it easier to understand.

Reviewer #4 (Recommendations for the authors):Overall, as it stands, this manuscript even if of high technological value, remains entirely descriptive and short in providing new insights into muscle structure and architecture. The main finding, an overlap between short Sls isoform and Proj in flight muscle sarcomeres, is redundant with the author's observation in the companion paper that in larval muscles expressing a long Sls isoform, Sls and Proj overlap as well.I would not recommend this manuscript for publication in eLife. It appears more suited for a specialized methodological development journal.Combination of Sls and Proj nanobodies with DNA-Paint represents an attractive example of technological development which alternatively could strengthen the co-submitted nanobodies toolkit manuscript.

Every structural paper reports the structure and is thus by definition descriptive. This is the aim of our manuscript. We do not think that the other nanobody resource paper reports an overlap of Sls and Projectin in the larvae. To resolve such a possible overlap, super resolution would be needed. The other paper does report that larval Sls isoform is dramatically stretched, more than 2 µm, and that Projectin is decorating the thick filament, likely in an oriented manner. If N-term of Projectin overlaps with C-term of Sallimus in this muscle is an open question that needs DNA-PAINT imaging of larval muscle. This requires a TIRF setting that is technically not trivial to achieve for larval muscle and hence has not been done by anybody.